# Intelligent Reflecting Surface-Based Non-LOS Human Activity Recognition for Next-Generation 6G-Enabled Healthcare System

**DOI:** 10.3390/s22197175

**Published:** 2022-09-21

**Authors:** Umer Saeed, Syed Aziz Shah, Muhammad Zakir Khan, Abdullah Alhumaidi Alotaibi, Turke Althobaiti, Naeem Ramzan, Qammer H. Abbasi

**Affiliations:** 1Research Centre for Intelligent Healthcare, Coventry University, Coventry CV1 5FB, UK; 2James Watt School of Engineering, University of Glasgow, Glasgow G12 8QQ, UK; 3Department of Science and Technology, College of Ranyah, Taif University, P.O. Box 11099, Taif 21944, Saudi Arabia; 4Faculty of Science, Northern Border University, Arar 91431, Saudi Arabia; 5School of Computing, Engineering and Physical Sciences, University of the West of Scotland, Paisely PA1 2BE, UK

**Keywords:** 6G, next-generation healthcare, intelligent reflecting surface, software-defined radio, RF sensing, machine learning

## Abstract

Human activity monitoring is a fascinating area of research to support autonomous living in the aged and disabled community. Cameras, sensors, wearables, and non-contact microwave sensing have all been suggested in the past as methods for identifying distinct human activities. Microwave sensing is an approach that has lately attracted much interest since it has the potential to address privacy problems caused by cameras and discomfort caused by wearables, especially in the healthcare domain. A fundamental drawback of the current microwave sensing methods such as radar is non-line-of-sight and multi-floor environments. They need precise and regulated conditions to detect activity with high precision. In this paper, we have utilised the publicly available online database based on the intelligent reflecting surface (IRS) system developed at the Communications, Sensing and Imaging group at the University of Glasgow, UK (references 39 and 40). The IRS system works better in the multi-floor and non-line-of-sight environments. This work for the first time uses algorithms such as support vector machine Bagging and Decision Tree on the publicly available IRS data and achieves better accuracy when a subset of the available data is considered along specific human activities. Additionally, the work also considers the processing time taken by the classier in training stage when exposed to the IRS data which was not previously explored.

## 1. Introduction

The wireless communication systems and specifically the small wireless devices used for wireless sensing in healthcare settings have experienced a trend toward achieving reliable detection. This includes obtaining high data rates, increased system capacity, higher carrier frequency, flexible/scalable hardware systems, and focusing energy radiation in an area of interest (beam-forming) [1]. Following the recent emergence of a fifth-generation communication system (5G), the next-generation wireless communication system (6G) is foreseen to exploit the intelligent reflecting surface (IRS) driven by cutting-edge deep learning and machine learning algorithms for energy optimisation, which will be one of the key components of future healthcare smart systems.

The IRS redirects the radio-frequency (RF) signals toward an area of interest and achieves beam-forming using a meta-surface that dynamically controls the optical and electrical properties, enabling the incident RF signals to steer towards an area of interest [2]. The traditional RF sensing techniques such as WiFi, radar, and software-defined radio (SDR) suffer from coarse-grain resolution and present low system accuracy when any monitored person performs activities of daily living at the corners of a room or in blind zones where RF signal reception is low or minimum. The beam-steering capability of IRS allows obtaining reliable and highly precise results in the case of activity recognition when performed in line-of-sight (LOS), non-LOS, and around corners as well. Due to the IRS’s substantial benefits in wireless communication, it has recently attracted a lot of research interest. The following are a few interesting applications of IRSs in wireless communication that were previously inconceivable [3,4,5,6].
The capacity of the IRS to increase coverage by establishing virtual connections for users with a blocked direct link to the base station is helpful for coverage extension, particularly in mm wave communication, which is severely impacted by blockages.An IRS can be placed at the cell edge to increase the required signal strength and minimise interference. Cell edge users suffer from both significant signal attenuation from their serving cell and severe co-channel interference from neighbouring cells.The wireless environment can efficiently manage the direction of user channel vectors by using an IRS. For instance, NOMA can be made viable by aligning the channels of two users.Data bits are contained in both the antenna index and the information symbol in index modulation. With a single RF chain and a low complexity transmitter, index modulation improves spectral efficiency [7,8].An IRS can be utilised as a hub for signal reflection to facilitate concurrent low power transmissions and interference reduction for large device-to-device communication.Furthermore, artificial multi-path propagation can be created by adding distributed IRSs to the LOS environment. As a result, the attainable rate is increased and spatial multiplexing is feasibly created.

### Machine Learning-Driven Intelligent Reflecting Surface

The machine learning-driven IRS consists of two layers, where the first layer is called the meta-surface, involving configurable unit cell elements that are intrinsically metal-o-dielectric and have sub-wavelength unit cell spacing. The second layer includes a network that primarily consists of the biasing and control components. The meta-surface layer essentially has a width of sub-wavelength and is electrically transversely big and intrinsically globally charged [9,10]. In comparison with other state-of-the-art relevant technologies such as relays and phased arrays, one of the major advantages of using IRS cells for RF signal beam-steering is its lower complexity, which makes the specific technology flexible and scalable to be deployed for larger areas at minimum power consumption cost.

The IRS measurement system works in conjunction with an SDR (universal software radio peripheral—USRP) that emulates a transmitter–receiver model presenting resolution beam-steering capacity and volume, specifically in the azimuthal plane [11,12]. The IRS system has demonstrated reliable results in terms of enhanced RF coverage area in indoor settings when tested for monitoring activities of daily living in non-LOS scenarios. The IRS measurement system has multiple connected columns having a near 3-bit phase resolution scheme integrated with three-pin diodes considering one unit copper cell area. The IRS measurement system is controlled through a WiFi network that enables it to be integrated into a 6G communication system with ease and comfort.

This work primarily presents a next-generation intelligent human activity recognition system that utilises an IRS, SDR, and machine learning algorithms where RF signals can be steered towards a particular area of interest by addressing the non-LOS communication issue encountered by present sensing technologies. We have monitored activities of daily living of different subjects including sitting down, standing up, and walking back and forth on multiple floors. Initially, the IRS measurement system was turned off and data were acquired for a specific set of activities; afterwards, the IRS system was turned on to acquire data for certain activities. A comparison of machine learning algorithm results is presented at the end of this paper that demonstrates the validity of the IRS by significantly improving the classification accuracy. A complete data flow diagram of the applied system in this study is demonstrated through Figure 1.

## 2. Literature Review

This section discusses the state-of-the-art literature on different contactless sensing approaches that have been effectively used in the past for irregularity detection including distinct human activity. The channel state information (CSI), radar, SDR, and received signal strength indicator (RSSI) are the prime approaches toward contactless sensing [13,14,15,16].

Radar-based sensing, which has a much wider bandwidth, is also used to detect human activities [17,18,19]. The frequency-modulated continuous wave (FMCW) radar uses a bandwidth up to 1.79 GHz, compared to WiFi technology which only utilises a bandwidth up to 20 MHz [20]. A higher spatial resolution of about 20 cm is provided by the micro-Doppler information extraction methods based on radar [21,22]. However, radar-based systems need specialised gear and computing power.

The CSI-based sensing that makes use of WiFi technology has lately gained popularity for feature extraction in the detection of several human activities [23]. A number of studies have concentrated on developing CSI-based applications, including those for detecting human presence [24], crowd counting [25], indoor setting localisation [26], and a fall or collapse detection system for the elderly [27]. Some recent research claims that WiFi signals can identify and differentiate the smallest motions made by a human body such as those made by the lips [28], the fingers on a keyboard [29], the heartbeat [30], and the rate of breathing [31].

The RSSI-based sensing for monitoring human activity is essentially reliant on differences in received signal intensity caused by different human activities [32]. The RSSI-based system’s detecting capability and accuracy are limited in comparison to SDR and CSI. A greater resolution for RSSI collection is made possible by the SDR-based technique, which increases identification accuracy by up to 72% [33]. The accuracy and area covered by RSSI-based systems are less due to the absence of frequency diversity seen in CSI-based systems. Orthogonal frequency division multiplexing (OFDM) is utilised to evaluate the CSI from each packet, whereas the RSSI is recorded as a single number per packet. As a result, when compared to RSSI, the CSI technique is more stable and offers more data. As a consequence, CSI is more resilient under challenging circumstances.

The SDR-based sensing uses hardware that was created especially to sense various human behaviours [34]. WiSee uses USRP to identify activities with up to 94% accuracy by detecting Doppler variations in wireless transmissions. Utilising specific circuit hardware, Allsee Technology developed a close-range sensing method for motion detection at distances of less than 2.5 feet. Utilising a platform based on SDR is an approach for eliminating radio CSI from WiFi signals without updating or altering the hardware [35,36].

## 3. Software-Defined Radio

The first communication system component for which hardware was enhanced with software was SDR. Reusing hardware components and giving components more flexibility were the two key ideas that were taken into consideration at the beginning of SDR. The first concept is shown by employing Viterbi codecs for channel and voice coding. The baseband processing where the properties of the transmitter and receiver can be modified instantly is an illustration of the second idea. This idea was first applied to software rather than hardware, but it now provides for a lot of flexibility in physical parts. In the context of 3G, when many frequency bands were allotted internationally, a serious demand for SDR emerged. SDR was utilised to reduce the number of possible baseband configurations [37].

Software is used to modulate and demodulate radio signals in the SDR communication system. The SDR is a general-purpose computer or a digital electronic device that can be reconfigured and performs a significant amount of signal processing. This idea proposes to create a radio that simply requires the installation of new software to receive and transmit a new radio protocol. SDR can be very useful for phone services that must provide a variety of radio protocols that are continually changing in real time, while digitised intermediate frequency signals are converted from and into an analogue form using analogue-to-digital and digital-to-analogue converters. The RF signals are transformed from and into analogue RF signals using a superheterodyne RF front end. The implementation of basic radio modem technologies is presently possible with SDR. The SDR is anticipated to eventually overtake all other radio communication technologies due to its obvious advantages [38]. The following are some fascinating capabilities of SDR that were not conceivable before.
SDR can change its configuration instantly, allowing the universal communication device to adapt to its surroundings. It could be a cordless phone one minute, a mobile phone the next, a wireless internet device the next, and a GPS receiver the next.Added functionalities can be rapidly and readily added to SDR. In actuality, the update might be sent wirelessly.Talking and listening to numerous channels are both possible on SDR.Radios that have never been made before can be created. Cognitive radios, sometimes known as smart radios, can analyse how the RF spectrum is being used in their local area and set themselves up for optimum performance.

## 4. Design of IRS-Based Measurement System

The IRS-based activity recognition system is designed and developed by the Communications, Sensing and Imaging group at the University of Glasgow, UK [39]. Figure 2, Figure 3, and Figure 4 are used from work in [39]. The IRS measurement system works based on unit cells and is depicted in Figure 2. The unit cell dimensions are shown in Table 1. The unit cells consist of five patches made up of copper material that is connected to a total of three pin diodes and a single capacitor. The five copper patches are etched on a dielectric substrate and are grounded for design purpose that has a relative permittivity of 2.65 and loss tangent value of 0.001. As shown in Figure 2, the copper patches are essentially wired to neighbouring cells at the upper side and lower sides to decrease the complexity of the IRS network with its configuration. The response function local reflection value against each cell element can be modified by changing the corresponding diode biasing states. The forward biased pin diode works as a tiny resistance and the reverse biased pin diodes function as a series capacitor considering two consecutive copper patches. A total of eight combinations of three-pin diodes are deployed, indicating binary values such as 000 to 111 as shown in Figure 2d.

The reversed-biased state is primarily obtained with 0 control voltage supplied and the forward-biased state is kept around 0.85 volts having a forward current of 3 milliamperes. The capacitor that operates at a self-resonant frequency gives a near-short circuit that performs tasks at a central frequency of 3.75 GHz and isolates the direct current bases RF signal paths. The copper path dimensions are optimised to greatly increase the resolution of the phase values of RF reflecting signals. In the IRS measurement system design for this work, a total number of seven different reflecting phase states are provided, having two out of the eight biasing contributions indicating a homogeneous value for response. The IRS measurement system for human activity recognition is shown in Figure 3 and involves an overall setup of 48 by 48 copper elements wired with a total number of 12 columns. The IRS measurement total dimensions used in this work and data acquisition are 1.08 m (width) and 0.72 m (height), at a central frequency of 3.75 GHz. The pin diodes on the copper cell elements obtain voltages (in power) from multiple shift registers at the rear side of the IRS elements. The IRS is controlled through a WiFi network connected to Raspberry Pi on a desktop computer system. The optimum configurations from the desktop computer are transmitted over a socket connection that transforms the values into a binary bit stream. Moreover, Table 2 provides essential parameters for the configuration of the USRP device.

## 5. Approach

### 5.1. Data Collection in a Multi-Floor Scenario

In this paper, we have used the publicly available online dataset that contains CSI of distinct human activities [40]. The data is collected using IRS system that is designed and developed by the Communications, Sensing and Imaging group at the University of Glasgow, UK. The data were collected in the experimental setup shown in Figure 4. Two participants, a female and a male, carried out three separate tasks: sitting, standing, and walking. It is important to understand that the actions of sitting and standing reflect the act of doing them, not the person’s posture or position while executing them. Furthermore, since the individuals were not required to keep their upper bodies motionless and immobile, there were modest variations in the upper body activity data for both sitting and standing.

The data were gathered in two phases for the multi-floor situation. In the first stage, both participants completed all tasks when the IRS was turned on, and then they repeated the tasks while the IRS was off. In total, 800 data samples were acquired including 50 data samples per activity. Similar to this, 1600 CSI samples were collected throughout each activity’s 4 s of operation. It is important to note that the acquired CSI values are complex numbers, which compromise both the amplitude and the phase data. These numbers are converted into comma-separated values (CSV) files and the amplitude information is extracted using a Python script. After data preparation, multiple machine learning algorithms are trained, validated, and tested using these CSV files.

In this research, we primarily took into account the signal amplitudes, which are shown in Figure 5 and Figure 6 for IRS-ON and IRS-OFF scenarios, respectively. These figures depict the CSI patterns of various body motions such as sitting, standing, and walking in a multi-floor situation. The 64 OFDM signal subcarriers are shown in each illustration using various colours. Each sub-figure’s x-axis displays the number of received packets, while the y-axis shows the amplitudes of subcarriers. In the situation of IRS-OFF, the received signal amplitude is insufficient to accurately differentiate between various operations. On the contrary, activating IRS results in noticeable modifications to the patterns of signals that are received, which are distinctive for various activities yet consistent across participants. Figure 5 shows a strong similarity between the sitting, standing, and walking actions of subject 1 and subject 2, which stimulates the suggested scheme’s generalisation to a wide variety of people.

### 5.2. Data Preprocessing and Machine Learning

Following data acquisition, when gathered data are saved in CSV files, it is common for part of the data to be missing owing to lost received packets, necessitating data equalisation. For data preparation and implementation of machine learning techniques, we utilised scikit-learn and NumPy, a popular Python data analysis toolbox. Additionally, CSV files are interpreted using the Python library Pandas. It converts CSV files into Python data frames for scikit-learn analysis. The data frame’s first column is where the labels are inserted. The dataset created by combining the data frames of each sample contains not-a-number (NaN) values as a result of a data length mismatch. Using the SimpleImputer built-in function of scikit-learn, these NaN values are replaced with the mean of each row. It is important to note that this kind of data equalisation has no impact on the general pattern of data. Following equalisation, these data were sent to three distinct machine learning algorithms: support vector machine (SVM), bagging classifier (BC), and decision tree (DT). These methods were selected after a thorough examination of several machine learning techniques on the dataset. Figure 1 depicts a complete block diagram of the proposed IRS-enabled human activity monitoring system.

Machine learning and deep learning approaches have been effectively used in the past for several applications including irregularity recognition [41,42,43]. In this study, we compared three distinct machine learning methods for the proposed IRS-enabled human activity monitoring system. The ability to accurately recognise various bodily motions is the assessment criterion taken into account. Accuracy was determined in two separate methods to conduct a full analysis. The first is K-fold cross-validation and the second is the test–train split. The K-fold cross-validation method, in which *k* is the number of groups into which a given data sample should be divided, is a well-known technique for assessing the effectiveness of a machine learning approach. The test–train split method is the second assessment strategy taken into account in this research. Using data that were not utilised to train the model, it produces predictions. This technique divides the dataset into two sections. The machine learning model is first applied to the training dataset as a portion of the data. The second portion of the dataset, which is used to assess performance, is the test dataset. In this study, 70% of the data are used for training, whereas 30% is used for testing.

## 6. Results and Discussion

After data preprocessing steps, the final datasets are fed into distinct machine learning algorithms for classification purposes. To attain reliable results, we performed simulations on three different well-known classifiers (SVM, BC, DT) and compared their performances. Figure 7 reveals classification report based on confusion matrix. As can be noted, it is a multi-class classification problem involving four distinct classes: sitting, standing, walking, and empty (vacant space with no activity). Figure 7a,c,e presents results of the IRS-OFF scenario and Figure 7b,d,f presents results of the IRS-ON scenario. The results in Figure 7a,b are based on SVM. Figure 7c,d results are based on BC. The results in Figure 7e,f are based on DT. The differences can be seen through the confusion matrix of the IRS-OFF scenario and IRS-ON scenario. The empty class attained almost the same accuracy in both scenarios as it is easy to classify by machine learning algorithms. However, activity classes such as sitting, standing, and walking were significantly improved in the IRS-ON scenario.

Table 3 presents results based on average training time in seconds taken by the algorithms and overall accuracy. We compared different algorithms’ performances for the IRS-OFF scenario and IRS-ON scenario. The training time of an algorithm primarily depends on three factors: the nature of the data, the structure of the algorithm, and the computational power of the system. Based on these factors, the training time of a machine learning model can be assessed. The simulations in this study were performed on a MacBook with an M2 chip, 8-core CPU/GPU, and 16-core neural engine. The same nature of numerical data stored in CSV files was fed into the considered machine learning algorithms and the training time was assessed. The training time of an algorithm can be higher or lower based on the above-mentioned three factors. As can be noted in Table 3, the training time of algorithms does not differ much in seconds, as there is an equal number of data points involved in both scenarios: IRS-ON and IRS-OFF. For the SVM classifier, the training time noted was 0.060 s for IRS-OFF scenario data and 0.075 s for IRS-ON scenario data. For the DT classifier, the training time was 0.140 s and 0.212 s for the IRS-OFF scenario and IRS-ON scenario, respectively. For the bagging classifier, the training time was reduced from 809.76 s to 607.79 s when trained on the IRS-ON scenario data. Overall, the BC attained more training time than SVM and DT due to its complex structure based on an ensemble meta-estimator.

In terms of classifier accuracy, it significantly improved through the IRS-ON scenario, as can be noted in Table 3. In the case of SVM, accuracy improved from 55% to 86%. In the case of BC, accuracy improved from 60% to 83%. In the case of DT, accuracy improved from 58% to 82%. Although SVM achieved the maximum accuracy of 86%, the rest of the algorithms also performed and improved quite significantly when the IRS was turned on. Based on these results, we can confidently state that IRS can be effectively utilised for multiple applications, especially activity sensing to attain accurate and reliable results. This study demonstrated a complicated multi-floor scenario of distinct activities that were successfully detected with the help of IRS technology.

### Challenges

There are certain challenges associated with IRS-assisted wireless communication that require further study. For instance, accurate phase shift tuning of every IRS element is required to ensure the effectiveness of IRS-aided wireless communication. This requires accurate channel estimation. The IRS’s limited capacity to send, receive and analyse signals makes it essentially passive in operation. To help users and base stations estimate channels, the IRS cannot emit pilot signals. The reflected signal on the IRS confronts an unfavourable route loss model, although the IRS controls the phases of dispersed waves to produce constructive interference at the target user. Moreover, a major overhead for the channel estimate procedure is also caused by the size of an IRS, which is typically large [44,45,46].

## 7. Conclusions

This study presents a novel approach to human activity monitoring such as sitting, standing and walking. An experiment has been carried out in a multi-floor scenario where a transmitter is positioned on one floor of the building and a receiver on another floor in order to create a challenging environment. The transmitter and receiver are connected to USRP devices in a non-LOS setting. A reconfigurable IRS is placed in between the transmitter and receiver. The above-mentioned three activities were carried out by two participants and data were collected while the IRS was off. Subsequently, data were collected while the IRS was turned on. The obtained data from both scenarios in the form of CSI was preprocessed and fed into machine learning algorithms. The results revealed that by turning on the IRS, three distinct machine learning algorithms, namely, SVM, BC, and DT, improved by 31%, 23%, and 24%, respectively. In conclusion, IRS technology has the ability to enable high precision and reliability for several applications including human activity monitoring even in non-LOS environments.

## Figures and Tables

**Figure 1 sensors-22-07175-f001:**
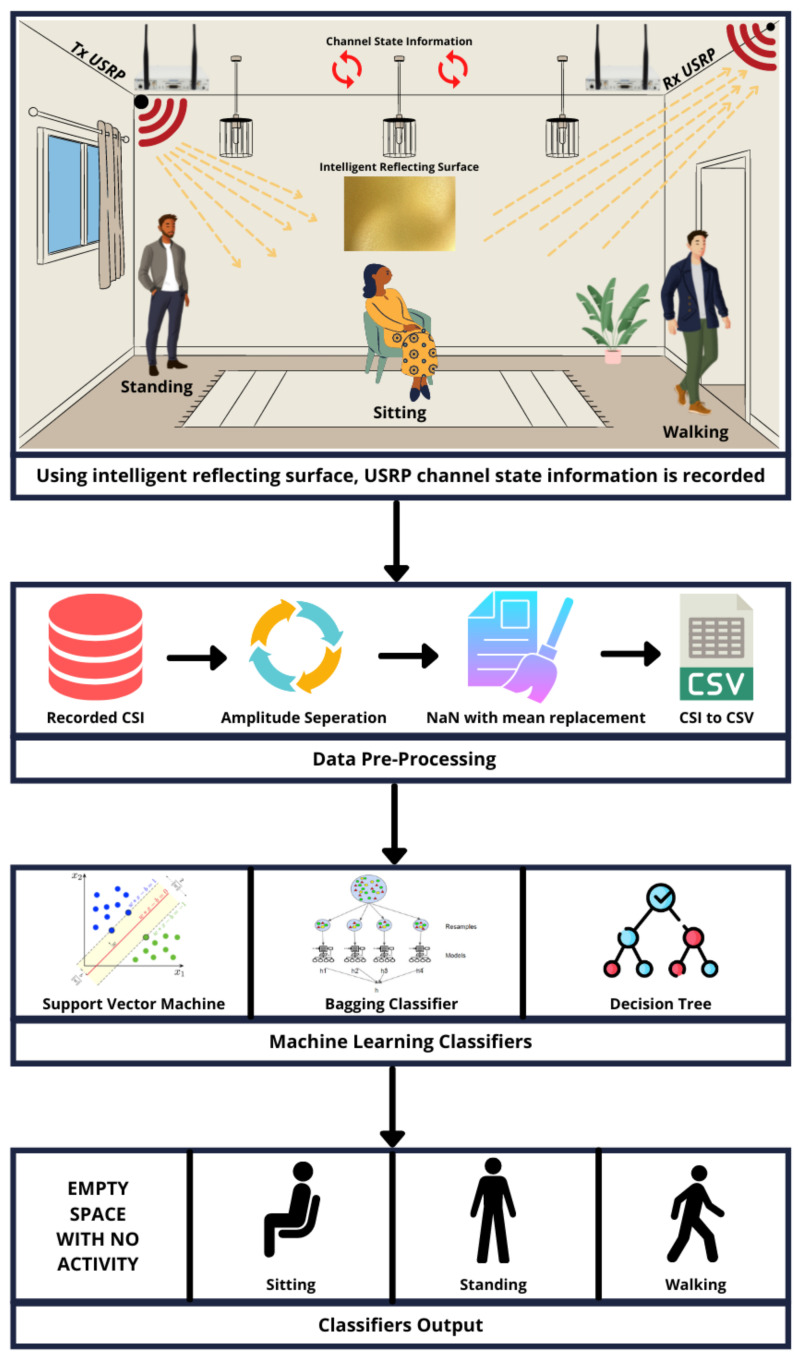
Block diagram of IRS-enabled system for distinct human activity monitoring.

**Figure 2 sensors-22-07175-f002:**
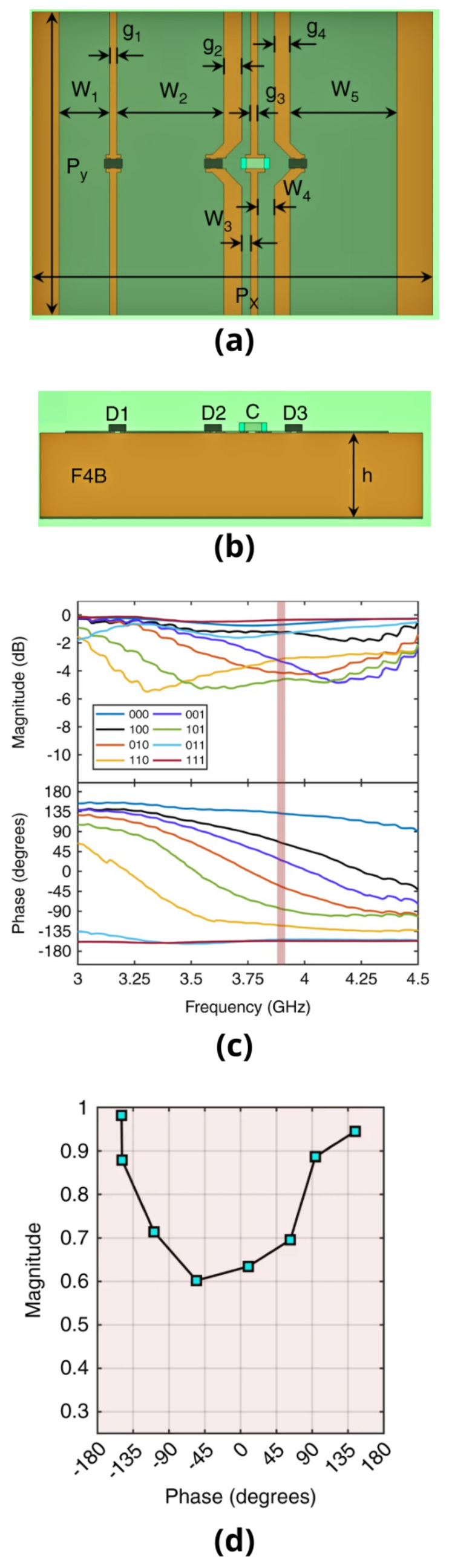
This work employed a unit cell design. Dimensions for the (**a**) front view and (**b**) cut view. The graph in (**c**) displays the deliberate reflection response as a frequency function for the eight-pin diode setups. The phase against magnitude plot at 3.9 GHz in graph (**d**) illustrates the IRS significant phase-dependent magnitude response [39].

**Figure 3 sensors-22-07175-f003:**
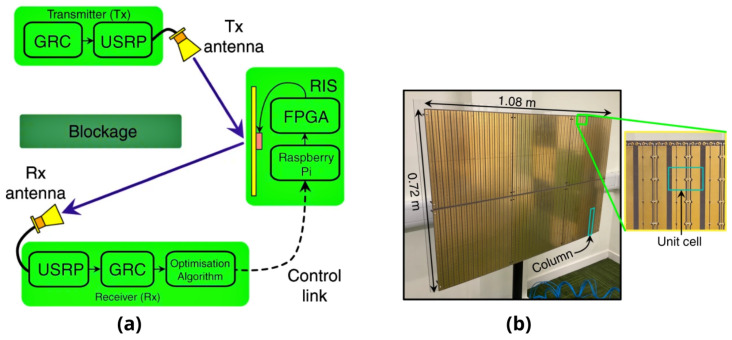
A wireless single-input, single-output communication technology supported by the IRS. (**a**) The IRS and Rx build a feedback loop to gradually increase the signal intensity from the Tx to the Rx antenna in order to overcome the obstacle. (**b**) A reconfigurable intelligent reflecting surface. The inset illustrates a column with choke inductors at the top. The 12 unit cells that make up the highlighted column are connected at the bottom and top by patches [39].

**Figure 4 sensors-22-07175-f004:**
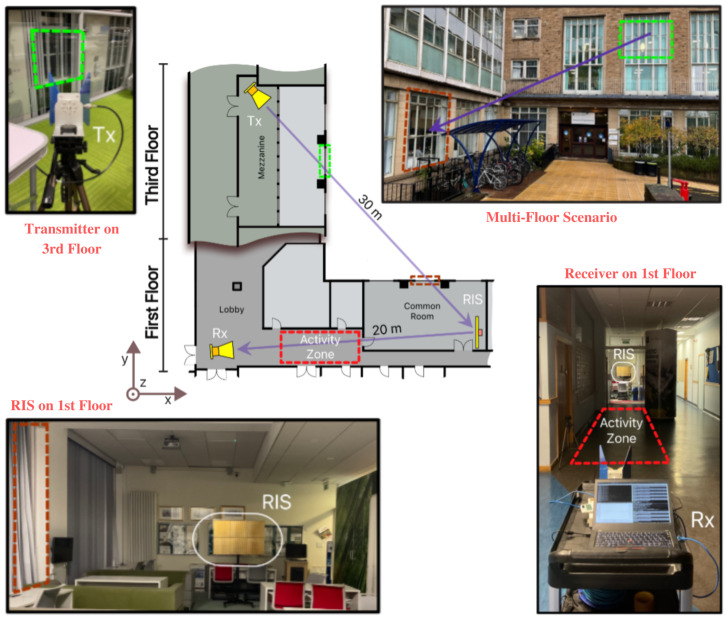
Setup for a multi-floor scenario experiment. IRS and Rx are located on the first floor, while the transmitter is situated on the third floor. Between IRS-Rx and Tx-IRS, a line-of-sight connection is created. In a hallway area between the receiver and the IRS, the activity zone is located [39].

**Figure 5 sensors-22-07175-f005:**
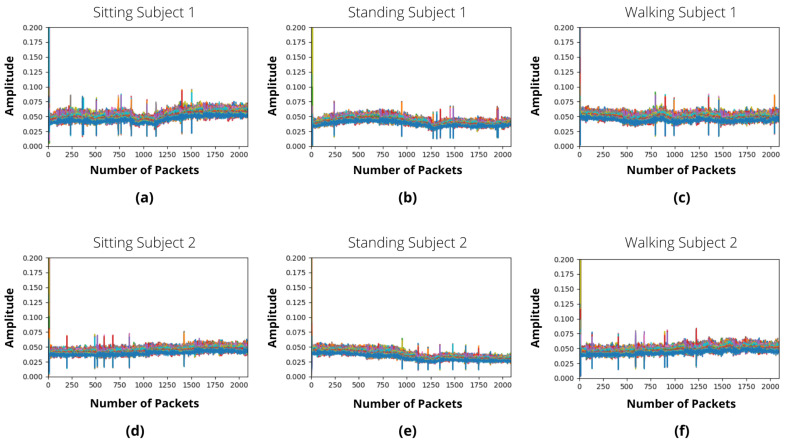
IRS-ON activities samples in form of CSI in a multi-floor scenario. (**a**) Person 1 sitting. (**b**) Person 1 standing. (**c**) Person 1 walking. (**d**) Person 2 sitting. (**e**) Person 2 standing. (**f**) Person 2 walking.

**Figure 6 sensors-22-07175-f006:**
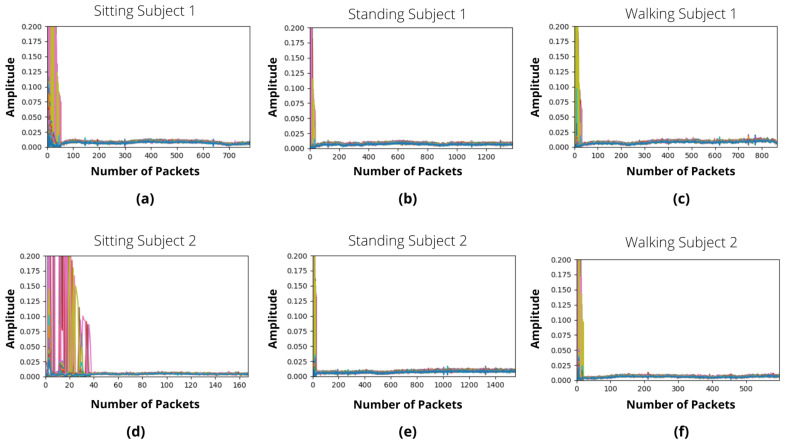
IRS-OFF activities samples in form of CSI in a multi-floor scenario. (**a**) Person 1 sitting. (**b**) Person 1 standing. (**c**) Person 1 walking. (**d**) Person 2 sitting. (**e**) Person 2 standing. (**f**) Person 2 walking.

**Figure 7 sensors-22-07175-f007:**
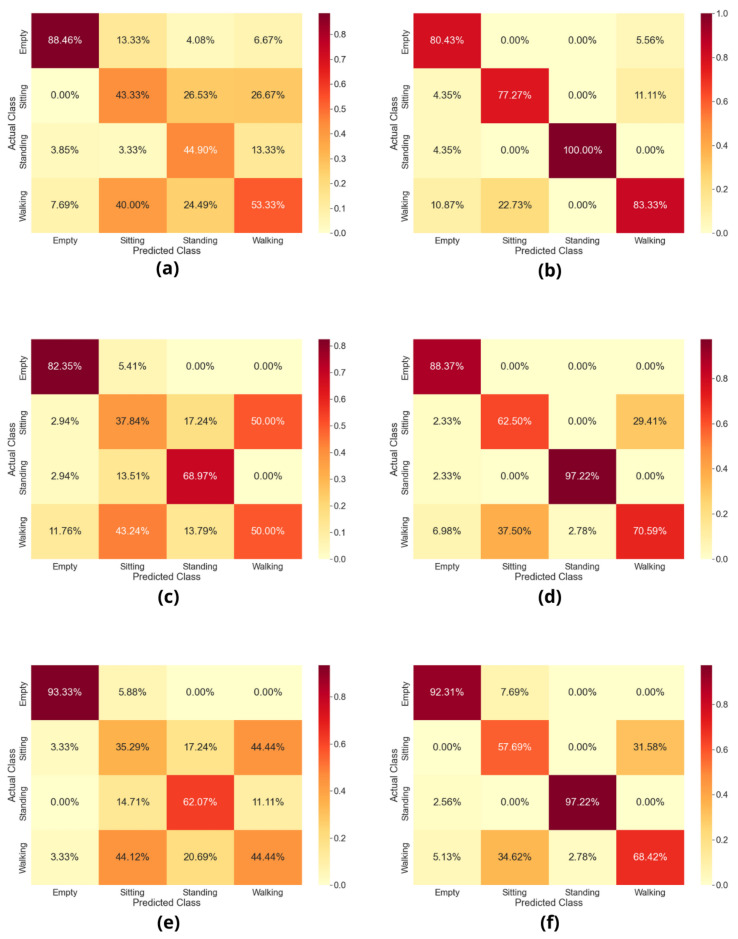
Classification results based on confusion matrix of multi-floor scenario. (**a**) SVM when IRS-OFF vs. (**b**) SVM when IRS-ON. (**c**) BC when IRS-OFF vs. (**d**) BC when IRS-ON. (**e**) DT when IRS-OFF vs. (**f**) DT when IRS-ON.

**Table 1 sensors-22-07175-t001:** Design dimensions for multi-bit unit cell.

Parameter	Periodicity, $P_{x}P_{y}$	Patch-Width, $W_{1}\;\mathrm{to}$ $W_{5}$	Patch-Spacing, $g_{1}\;\mathrm{to}$ $g_{4}$	Substrate Thickness, *h*
Dimensions in mm	22.5; 15.0	6.0; 0.9; 0.5; 6.0; 2.9	0.9; 0.4; 1.0; 0.4	5.0

**Table 2 sensors-22-07175-t002:** Parameters for configuring USRP hardware and software.

Parameter	Value
OFDM Subcarrier	64 Carriers
Bit Per Symbol	2 Bits
Pilot Subcarrier	4
Devices Used	USRP X300/USRP X310
Channel Mapping	1 Tx, 2 Rx
Central Frequency	3.75 GHz
Data Type	Int16
Gain (dB)	Tx 10, Rx 2

**Table 3 sensors-22-07175-t003:** Machine leaning classifiers average training time in seconds and overall accuracy score on IRS-OFF and IRS-ON scenarios.

Classifier	IRS-OFF Scenario		IRS-ON Scenario	
	Training Time	Accuracy	Training Time	Accuracy
Support Vector Machine	0.060 s	55%	0.075 s	86%
Bagging	809.76 s	60%	607.79 s	83%
Decision Tree	0.140 s	58%	0.212 s	82%

## Data Availability

Not applicable.

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
