# Peer review of "Intelligent Reflecting Surface-Based Non-LOS Human Activity Recognition for Next-Generation 6G-Enabled Healthcare System"

_sensors, 2022, doi:10.3390/s22197175_

Round 1

Reviewer 1 Report

This paper investigates the intelligent reflecting surface (IRS) to assure high accuracy activity monitoring in complicated environments where traditional microwave sensing is ineffective. The critical aspects, from my point of view, are the following ones:

Problem 1: In the Reference, even if there is not a lot of recent interest in this topic, the authors could make clear why such type of research is important for academics in Introduction.

Problem 2The quality of the Figure should be carefully revised to improve the resolution.

Problem 3As shown in Table 3, the overall accuracy significantly improved through the IRS-ON scenario. However, the training time is also increased in some cases. More detailed interpretation should be given.

Problem 4: As shown in Section 6, the accuracy and training time is considered to verified the characteristic of the proposed scheme. However, this is not enough to verify the effectiveness of the proposed scheme.

Problem 5What are the disadvantages of your proposed methods?

Round 2

Reviewer 1 Report

The manuscript has been comprehensively revised according to the reviewer’s comments, and it can be accepted in present form.

Reviewer 2 Report

The authors have addressed my concerns, no further comments.